# Identification of Multimodal Stance Towards Frames of Communication

**Maxwell A. Weinzierl** and **Sanda M. Harabagiu**

Human Language Technology Research Institute, The University of Texas at Dallas

{maxwell.weinzierl, sanda}@utdallas.edu

## Abstract

Frames of communication are often evoked in multimedia documents. When an author decides to add an image to a text, one or both of the modalities may evoke a communication frame. Moreover, when evoking the frame, the author also conveys her/his stance towards the frame. Until now, determining if the author is in favor of, against or has no stance towards the frame was performed automatically only when processing texts. This is due to the absence of stance annotations on multimedia documents. In this paper we introduce MMVAX-STANCE, a dataset of 11,300 multimedia documents retrieved from social media, which have stance annotations towards 113 different frames of communication. This dataset allowed us to experiment with several models of multimedia stance detection, which revealed important interactions between texts and images in the inference of stance towards communication frames. When inferring the text/image relations, a set of 46,606 synthetic examples of multimodal documents with known stance was generated. This greatly impacted the quality of identifying multimedia stance, yielding an improvement of 20% in F1-score.

## 1 Introduction

Frames of communication select particular aspects of an issue and make them salient in communicating a message (Entman, 1993). For example, when discussing the confidence in the COVID-19 vaccine, selecting the aspects concerning the development of the vaccine may convey the message that the vaccine is safe because (a) scientists have worked for decades on coronavirus vaccines, and (b) the vaccine has been tested and tracked, as shown in the frame illustrated in Figure 1. Frames of communication are ubiquitous in social media discourse and can impact how people understand issues and, more importantly, how they form their opinions (Chong and Druckman, 2007). Previous

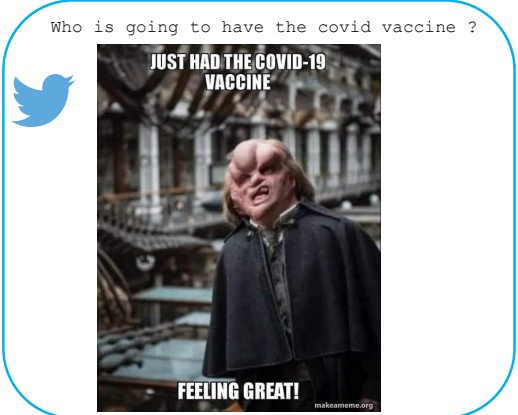

Figure 1: Frame evoked in multimedia document.

computational methods (Card et al., 2016; Hartmann et al., 2019; Mendelsohn et al., 2021) have focused on the problem of discovery of communication frames, or more precisely, identifying when a frame is evoked in a text. However, frames are also evoked in images, not only in texts, as advocated in Entman (2003), as text/images combinations convey memorable, emotionally-charged messages.

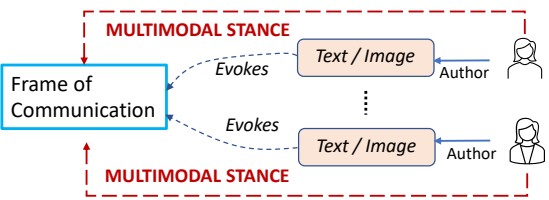

Figure 2: Communication frame evoked in multimedia.

In documents where a frame of communication is evoked, the authors of the documents also express their stance towards the frame. This allows us to consider the *multimodal stance* towards frames as an extension of the original notion of frame evocation introduced in Entman (2003), as shown in Figure 2.

Figure 1 illustrates a frame evoked in a tweet. Understanding why the frame is evoked requires accounting for the interaction between the tweet text and the image. While the question from the text of Figure 1 does not necessarily articulate the problem of vaccine safety, the image shows the result of vaccination, and the text superposed on the image is sarcastic. Together, the text and the image evoke the frame because they address the aspects highlighted by the frame, referring to confidence in the COVID-19 vaccine. Moreover, the tweet author also conveys her/his *stance* towards the frame, namely that they disagree with it.

The goal of stance detection is to determine whether the author of a document is in favor of, against, or has no stance toward a specific target (Mohammad et al., 2016; Hardalov et al., 2022). Instead of considering stance towards a target, we consider stance towards a frame of communication, which articulates precisely the relevant aspects highlighted by the frame. In this case, the stance is judged against *explicit* aspects of the frame, not against some unspecified properties of a target. Moreover, since until now stance identification methods relied only on texts, we propose to also consider images and their interaction with texts in the determination of the stance of a document towards a frame of communication. To be able to develop computational methods capable of inferring the stance of the author of a multimedia document towards a frame of communication, we need to rely on annotated examples. To our knowledge, there are no such annotations currently available.

In this paper we introduce a new dataset of Twitter postings which (1) contain a combination of text and image(s) that (2) evoke a frame of communication, annotating (3) the stance towards the frame of communication expressed by the tweet author. For this purpose, we have considered 113 different frames of communication related to COVID-19 vaccination, described in Section 2, which were evoked across 11,300 such tweets. The resulting MMVAX-STANCE dataset enabled us to experiment with several stance detection methods to ascertain a baseline performance for this specific task. To further improve the performance, we employ self-training that models the interactions between text and images. We also employ stance detection operating only on text which was trained on a separate, large dataset annotated with stance information. This allowed us to consider several data augmentation methods, which proved to improve the results of multimedia stance detection. The MMVAX-STANCE dataset, code, and other resources are available on GitHub[1].

The remainder of the paper is organized as follows. Section 2 presents the communication frames that were used in the multimodal annotation of stance. Section 3 describes the annotations, while models used for multimodal stance detection are presented in Section 4. Section 5 outlines the experimental results, which are also discussed. Section 6 describes the related work, while Section 7 summarizes the conclusions.

## 2 Frames of Vaccine Hesitancy

As reported in Glandt et al. (2021), the recent COVID-19 pandemic has given rise to a large variety of opinions regarding the public health mandates designed to contain the spread of the virus. While a stance-annotated dataset including tweets targeting these measures was introduced in Glandt et al. (2021), no opinions determined by the emergence of the COVID-19 vaccines were captured. These vaccines were hailed with enthusiasm by many, but substantial opposition and hesitancy towards vaccines also emerged on social media. This motivated our interest in capturing the stance towards multiple frames of communication that highlight aspects of vaccination.

Attitudes towards vaccination, e.g. vaccine readiness or hesitancy, are modeled by a set of seven components or factors that increase or decrease an individual's likelihood of getting vaccinated (Geiger et al., 2022). These components are listed in Table 1 along with their definitions and examples of frames of communications that address aspects of each component. We note that the vaccination components listed in Table 1 can be interpreted as dimensions of the frames, similar to the frame dimensions introduced in the Media Frames Corpus (Card et al., 2016).

In our work, we have considered the same 113 frames of communication that were reported in Weinzierl and Harabagiu (2022c), resulting from complex answers to questions that addressed various aspects of each of the components used in the 7C model (Geiger et al., 2022). The majority of the 113 frames that we considered address more than one of the components listed in Table 1. A total of

---

[1]https://github.com/Supermaxman/MMVax-Stance

| Component | Definition | Examples of Frames of Communication |
|---|---|---|
| Confidence | Trust in the security and effectiveness of vaccinations, the health authorities, and the health officials who recommend and develop vaccines. | ☐ Pfizer COVID-19 vaccine may cause anaphylaxis in people with polyethylene glycol (PEG) allergy. ☐ The Government has provided plenty of safety information about the COVID-19 vaccines. |
| Complacency | Complacency and laziness to get vaccinated due to low perceived risk of infections. | ☐ Preference for getting COVID-19 and fighting it off than vaccinating. |
| Constraints | Structural or psychological hurdles that make vaccination difficult or costly. | ☐ It takes courage both to vaccinate against COVID-19 and to refuse the vaccine. |
| Calculation | Degree to which personal costs and benefits of vaccination are weighted. | ☐ COVID-19 vaccines protect against the emerging variants. |
| Collective Responsibility | Willingness to protect others and to eliminate infectious diseases. | ☐ Vaccination is key in protecting yourself and others against COVID-19. |
| Compliance | Support for societal monitoring and sanctioning of people who are not vaccinated. | ☐ People choosing not to get the COVID-19 vaccine should not lose venue access/travel to some countries. |
| Conspiracy | Conspiracy thinking and belief in fake news related to vaccination. | ☐ COVID-19 vaccines make you 5G compatible. ☐ The COVID vaccine renders pregnancies risky. |

Table 1: Components of Vaccination Hesitancy and Examples of Frames of Communication addressing them in MMVAX-STANCE.

57 frames address Confidence in vaccines (53%); 43 frames address Calculation (43%); 37 of the frames address Conspiracy (33%); 28 of the frames address Compliance (23%); 14 frames address Collective Responsibility (12%); 11 frames address Complacency (10%) and only 7 frames address the Constraints (6%). Moreover, the large number of frames of communication that we have considered sets apart our multimodal stance-annotation effort when compared with other existing stance detection annotated datasets using a much smaller number of targets.

## 3 The MMVAX-STANCE Dataset

**Data collection.** After approval from the University of Texas at Dallas Institutional Review Board was obtained, (IRB-21-515 stipulated that our research met the criteria for exemption #8(iii) of Chapter 45 of Federal Regulations Part 46.101.(b)), we used the Twitter historical API with the following query: [(covid OR coronavirus) vaccine lang:en], retrieving 33,566,030 original tweets from December 18th, 2019, to January 1st, 2022 (approx. 2 years). 1,920,923 of these tweets (6%) contained not only text, but also images, thus they are multimodal documents. 75% of multimodal tweets had one image, while 14% had two images, and 11% had three or more images. A large fraction of these multimodal tweets were duplicates, which required filtering. Perceptual Hashing (pHash) (Zauner, 2010) is a commonly used technique for image duplication detection on a massive collection of images. We utilized pHash to remove duplicate images, and their corresponding tweets, resulting in

a final collection of 1,099,645 multimodal tweets.

**Data selection.** To identify from the collection of 1,099,645 tweets those that are relevant to any of the 113 frames of communications described in Section 2, we relied on multiple retrieval systems which employ dense indices utilizing CLIP (Radford et al., 2021) with FAISS (Johnson et al., 2021). CLIP is a powerful image-text encoder, which was trained on image-caption pairs to produce embeddings that are close in distance if both the caption and image are aligned semantically. Index $I_{\mathcal{I}}$ was constructed using CLIP's image encoder, which produced embeddings of size 768 for each image in the collection of 1,099,645 multimodal tweets. Index $I_{\mathcal{T}}$ was constructed using CLIP's text encoder, which produced embeddings of size 768 for the text of each tweet. Index $I_{\mathcal{J}}$ was constructed using both of CLIP's encoders, where the centroid embedding of size 768 was produced from each tweet's text embedding and image embedding. Each frame of communication was also embedded using each of the above three approaches, and these frame embeddings were used to query all three dense indices for relevant tweets by finding the closest embeddings by distance. The three indexes informed our selection of the data in the following way: The top 100 most relevant tweets were selected across all three indices for each frame of communication, producing a total of 11,300 tweets to be annotated. Table 2 presents the top 15 most common tokens present in these tweets, along with the top 15 most common hashtags, mentioned users, and linked URL domains. Tokenization was performed with spaCy (Honnibal et al., 2020), with $31 \pm 18$ tokens discovered per tweet.

| | Tokens | vaccine / vaccinate / vaccinated, COVID / COVID-19, get / getting, people, immunity, virus / coronavirus, children, still, Pfizer, effective, need, know, MRNA, read, new |
|---|---|---|
| | Hashtags | #covid19 / #covid / #covid_19, #vaccine / #vaccines / #covidvaccine / #vaccination, #coronavirus, #pfizer, #pandemic, #mrna, #health, #astrazeneca, #vaccineswork, #deltavariant |
| | Mentions | @who, @cdcgov, @us_fda, @potus, @pfizer, @realdonaldtrump, @rwmalonemd, @nytimes, @bharatbiotech, @cdcdirector, @joerogan, @cnn, @fda, @ocugen, @randpaul |
| | Domains | cdc.gov, medrxiv.org, nebraskamed.com, fda.gov, gov.uk, health.gov.au, youtube.com, nytimes.com, timesofindia.indiatimes.com, nhs.uk, nature.com, who.int, livemint.com, theguardian.com, tga.gov.au |

Table 2: Top 15 most common tokens, hashtags, mentioned users, and linked URL domains in the tweets from MMVAX-STANCE.

| | Accept | Reject | No Stance | Not Relevant | Total | Tweets |
|---|---|---|---|---|---|---|
| Train | 2,371 | 1,236 | 1,857 | 2,448 | 7,912 | 4,882 |
| Dev | 196 | 117 | 333 | 397 | 1,043 | 646 |
| Test | 578 | 332 | 642 | 793 | 2,345 | 1,390 |
| Total | 3,145 | 1,685 | 2,832 | 3,638 | 11,300 | 6,918 |

Table 3: Distribution of stance values and unique tweets for each split in MMVAX-STANCE.

**Data annotation.** First, researchers from The University of Texas at Dallas judged that 7,662 of the retrieved multimodal tweets evoke the corresponding frames of communication. The Cohen's Kappa score that we obtained for inter-annotator agreement was 0.82. Each tweet was labeled with stance values by three researchers from The University of Texas at Dallas. The questionnaire presented to each annotator, detailed in Appendix A, allowed us to annotate each tweet that evoked a frame of communication with one of the stance values: *Accept*, *Reject*, or *No Stance*. The results were: 3,145 tweets which *Accept* their frame, 1,685 which *Reject* their frame, and 2,832 with *No Stance*. The Cohen's Kappa score for stance inter-annotator agreement was 0.69.

**Benchmark subsets.** Tweets were further separated into training, development, and test collections to foster experimental reproducibility, outlined in Table 3. Care was taken to ensure each collection had entirely unique tweets, such that there was no overlap between training, development, and testing. The training collection, which consists of 7,912 multimodal stance judgments, was utilized to train our automatic multimodal stance identification systems, described in Section 4. The development collection, which consists of 1,043 multimodal stance judgments, was used to select system hyperparameters. The test collection, which consists of 2,345 multimodal stance judgments, was used to evaluate the several stance identification approaches, enabling us to experiment with MMVAX-

STANCE and report the results in Section 5.

## 4 Multimodal Models of Stance

### 4.1 Baseline Models

Several models were used to establish baseline results on MMVAX-STANCE. Transformer-based pre-trained language models such as Bidirectional Encoder Representations from Transformers (BERT) (Devlin et al., 2019) are commonly utilized for text-based stance identification (Hossain et al., 2020; Weinzierl et al., 2021; Weinzierl and Harabagiu, 2022b; Barbieri et al., 2020). Each tweet's text is provided along with the text of the communication frame, and a softmax layer is added on top of the contextualized "[CLS]" embedding to perform stance identification. Improvements to text-based stance identification have been found through combining BERT with lexical, emotional, and semantic Graph Attention Networks (Veličković et al., 2018; Weinzierl et al., 2021) and Moral Foundation (MF) Hopfield pooling (Ramsauer et al., 2021; Weinzierl and Harabagiu, 2022c), referred to as the LES-GAT-MF system. Text-based stance identification can also benefit from exploiting implicit attitude consistency relationships between agreeing and disagreeing tweets (Weinzierl and Harabagiu, 2022a). This approach is referred to as the LACRscore system. Each of the text-based stance identification methods was image-informed by providing a textual description of images. We considered two methods: Optical Character Recognition (OCR) and image captioning (Caps). OCR

was performed utilizing the Tesseract OCR engine (Smith, 2007), while image captioning was performed by BLIP-2 (Li et al., 2023).

CLIP was also considered, which was trained to produce aligned image/text embeddings through contrastive learning with a text-based encoder and a separate image-based encoder. CLIP-Text utilized the text encoder of CLIP, and therefore it did not have access to any image information for each tweet. Both the tweet's text and the communication frame's text would be encoded through CLIP's text encoder, with both embeddings concatenated together. CLIP-Image similarly utilized the image encoder of CLIP, and therefore did not have access to any text information for each tweet. CLIP-Joint utilized both the text and the image encoder to produce a CLIP text embedding and a CLIP image embedding for each tweet, which were concatenated together. A final softmax layer was added to the concatenated embeddings to perform stance identification for each CLIP-based method.

Multimodal transformer methods were considered for stance identification, as a fusion of context across both text and image modalities could be necessary to fully understand the stance of a tweet. VILT (Kim et al., 2021) is a Vision-and-Language Transformer that models interactions between the language in text and the visual components in an image using a cross-attention transformer encoder architecture. FLAVA (Singh et al., 2022) is a Foundational Language and Vision Alignment Model, which was pre-trained on about 7 times more data than VILT, and also included additional contrastive and unimodal pre-training tasks. BridgeTower (BT) (Xu et al., 2023) bridges the gap between text and image encoders by introducing a cross-modal encoder that attends to various levels of the text and image encoders, enabling the text encoder to be modeled differently than the image encoder, while the cross-modal encoder can benefit from the specialization of both encoders. Each multimodal system encodes the tweet's text, the communication frame, and the tweet's image and produces a single contextualized embedding, which is provided to a softmax layer that performs stance identification.

## 4.2 Accounting for Text-Image Relations

As we have seen in the example illustrated in Figure 1, the stance of a multimodal document towards a frame of communication depends on the *meaning* derived from the interaction between the textual part and the image of the document. Multiple types of relations between texts and images appearing in the context of the same document have been covered in Bateman (2014). For the inference of the stance of the author of a multimedia document towards a communication frame we have considered whether the text and the image pull in the same direction of the stance, thus they are *convergent*, or *divergent*. According to the distinction proposed in Kloepfer (1976), convergent relations between texts and images can be further separated as *parallel*, when the text and the image convey the same stance towards the frame, or *additive*, when the meaning of one of the modalities adds to the meaning of the other in determining the stance.

We hypothesized that relations between the text and the image of each multimodal tweet can be inferred if we compare the annotated multimodal stance with the stance inferred only from the text. As currently, there are many text-based stance detection models that operate quite well, we were able to produce silver annotations of the text-based stance for each example from MMVAX-STANCE. For this purpose, we have used the text-based stance detection reported in Weinzierl and Harabagiu (2022b). This allowed us to establish seven different prototypical examples which combine the values of the annotated multimedia stance with the values of the text-based stance annotations. The prototypical examples are illustrated in Figure 3. We found that 1,578 of the examples from the training set of MMVAX-STANCE represent prototype 1; 249 examples represent prototype 2; 684 examples represent prototype 3; 249 examples represent prototype 4; 1,052 examples represent prototype 5; 303 examples represent prototype 6; and 544 examples represent prototype 7.

**Inference of Text/Image Relations:** The prototypical examples inform the possible relations between the text and the image of each multimodal tweet, which in turn help us derive the possible stance of the image.
□ In Prototype 1 and Prototype 3, as seen in Figure 3, the multimedia stance and the text-only stance have the same value: *Accept* for Prototype 1 and *Reject* for Prototype 3. This indicates that there are two possible relations between the text and the image in each of these prototypes: either the text and the image have a *Parallel* relation, each contributing separately to the value of the multimodal stance; or the text and the image share a *Diverging*

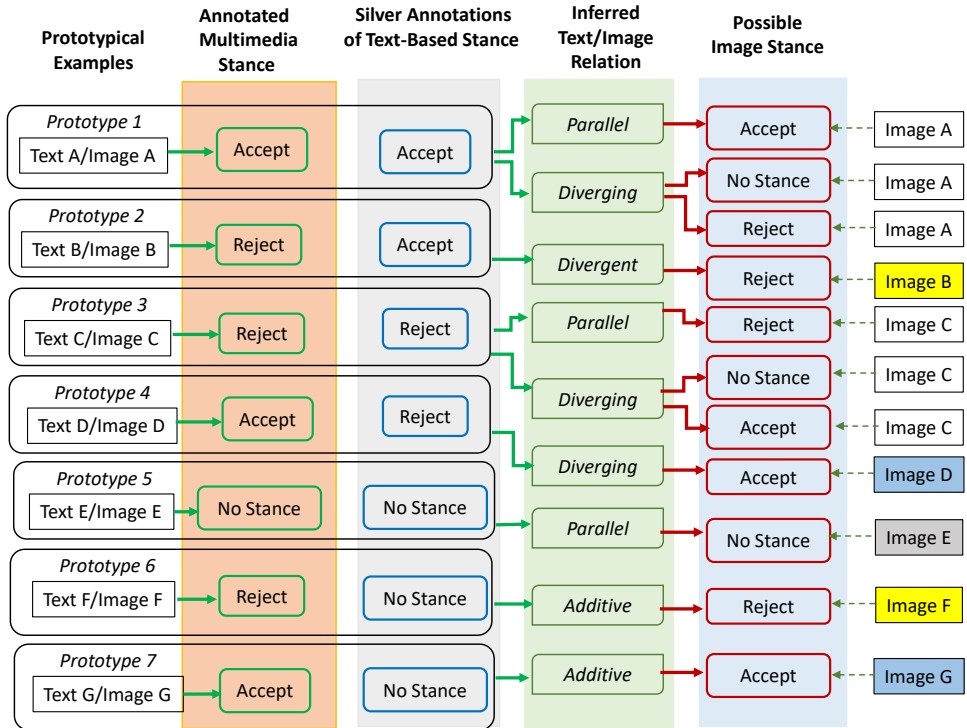

Figure 3: Inference of text/image relations enables deriving the possible image stance values.

relation, in which the image has a stance of a different value towards the frame than the text. When a *Parallel* relation is inferred, the possible stance of Image A is only *Accept* and the possible stance of Image C is *Reject*. But when a *Diverging* relation is inferred, the stance of Image A, or Image C, may be either *No stance* or *Reject*.

☐ In Prototype 2 and Prototype 4, the stance of the multimedia document towards the frame of communication is different from the stance predicted for the text, indicating that the only possible relation between text and image is *Diverging*. Therefore, the possible stance value for the image is the same as the multimedia stance value.

☐ In Prototype 5, because both the multimedia stance and the text stance have the value *No Stance*, this enables us to infer that the text and the image share a *Parallel* relation. Therefore, the only possible value for the stance of Image E is *No stance*.

☐ In Prototype 6 and Prototype 7, because the stance value of the text is *No Stance*, the relation between Text F and Image F, or between Text G and Image G, is *Additive*. This entails that Image F can have only a stance value of *Reject*, while Image G can have only a stance value of *Accept*.

To create a large set of synthetic examples, we considered the interplay between *Parallel*, *Additive*, and *Diverging* relations between texts and images.

Hence, all images from tweets only having *Accept* possible stance values, due to *Additive* or *Parallel* relations, were paired with texts from tweets having *Accept* text-based stance values from *Parallel* relations. In the same way, all images having only *Reject* possible stance values, due to *Additive* or *Parallel* relations, were paired with texts from tweets that were predicted to have *Reject* stance values from *Parallel* relations.

**Generating Synthetic Multimedia Stance Examples:** Figure 3 shows that there is one prototype for which the text stance is deemed *Accept* and the relationship between the text and the image of the tweet could be *Parallel*: Prototype 1. There are also two prototypes that have possible image stance value only as *Accept*: Image D from Prototype 4, due to a *Diverging* relationship, and Image G from Prototype 7, due to an *Additive* relationship. If we pair all the texts from Prototype 1 with all the images from the examples of Prototype 7, we generate 8,371 synthetic examples with a multimodal stance *Accept*. Similarly, if we pair all the texts from Prototype 1 with all the images from the examples of Prototype 4, we generate 3,732 synthetic examples with a multimodal stance *Accept*. In this way, we have created 12,103 synthetic multimodal documents with the multimodal stance *Accept*, which we refer to as $S_{Accept}$. Figure 3 also

| System | Macro $F_1$ | Macro P | Macro R | Accept $F_1$ | Accept P | Accept R | Reject $F_1$ | Reject P | Reject R |
|---|---|---|---|---|---|---|---|---|---|
| BERT | 37.69 | 37.67 | 37.88 | 45.81 | 44.02 | 47.75 | 29.57 | 31.31 | 28.01 |
| BERT + OCR | 37.45 | 39.84 | 35.66 | 45.98 | 45.94 | 46.02 | 28.92 | 33.73 | 25.30 |
| BERT + Caps | 44.07 | 44.90 | 43.51 | 50.55 | 49.10 | 52.08 | 37.60 | 40.70 | 34.94 |
| BERT + OCR + Caps | 40.30 | 39.54 | 41.49 | 50.48 | 46.88 | 54.67 | 30.13 | 32.19 | 28.31 |
| LES-GAT-MF | 39.02 | 39.67 | 38.45 | 46.17 | 45.75 | 46.61 | 31.86 | 33.60 | 30.29 |
| LES-GAT-MF + OCR | 38.40 | 39.11 | 37.91 | 45.38 | 44.18 | 46.65 | 31.43 | 34.05 | 29.18 |
| LES-GAT-MF + Caps | 42.66 | 42.13 | 43.22 | 48.21 | 46.96 | 49.53 | 37.11 | 37.31 | 36.92 |
| LES-GAT-MF + OCR + Caps | 42.26 | 42.10 | 42.46 | 49.13 | 48.09 | 50.22 | 35.39 | 36.11 | 34.70 |
| LACRscore | 41.74 | 40.39 | 43.22 | 47.21 | 46.35 | 48.10 | 36.27 | 34.42 | 38.34 |
| LACRscore + OCR | 41.27 | 39.65 | 43.06 | 46.08 | 44.83 | 47.41 | 36.46 | 34.46 | 38.70 |
| LACRscore + Caps | 44.40 | 43.34 | 45.52 | 50.09 | 49.27 | 50.93 | 38.71 | 37.41 | 40.11 |
| LACRscore + OCR + Caps | 43.96 | 43.32 | 44.66 | 48.81 | 47.31 | 50.41 | 39.11 | 39.32 | 38.91 |
| CLIP-Text | 39.70 | 39.33 | 40.25 | 47.06 | 44.87 | 49.47 | 32.34 | 33.78 | 31.02 |
| CLIP-Image | 28.67 | 33.57 | 50.13 | 53.93 | 37.14 | **98.44** | 3.41 | 30.00 | 1.81 |
| CLIP-Joint | 46.79 | 43.42 | 53.78 | 56.29 | 45.60 | 73.53 | 37.29 | 41.24 | 34.04 |
| VILT | 44.37 | 38.62 | 52.52 | 53.52 | 48.39 | 59.86 | 35.21 | 28.85 | 45.18 |
| FLAVA | 47.63 | 42.93 | 53.52 | 55.76 | 49.60 | 63.67 | 39.51 | 36.27 | 43.37 |
| BT | 52.91 | 46.84 | 60.86 | 61.74 | 53.87 | 72.32 | 44.09 | 39.81 | 49.40 |
| BT + $S_{All}$ - $D_T$ | 54.46 | 50.79 | 58.73 | 59.78 | 55.29 | 65.05 | 49.15 | 46.28 | 52.41 |
| BT + $S_{Same}$ | 57.22 | 53.12 | 62.03 | 62.30 | 57.32 | 68.23 | 52.14 | 48.92 | 55.82 |
| BT + $S_{Accept}$ | 60.65 | 57.74 | 63.92 | 69.27 | 65.03 | 74.11 | 52.03 | 50.44 | 53.72 |
| BT + $S_{Reject}$ | 62.10 | 59.78 | 64.70 | 65.16 | 61.29 | 69.55 | 59.04 | 58.26 | 59.85 |
| BT + $S_{All}$ | **71.32** | **71.51** | **71.16** | **79.45** | **78.64** | 80.28 | **63.19** | **64.38** | **62.05** |

Table 4: Results from the stance identification experiments on the test collection from MMVAX-STANCE.

shows that there is one prototype for which the text stance *Reject* is held towards a frame of communication, and the relationship between the text and the image of the tweet could be *Parallel*: Prototype 3. Image B used in Prototype 2 and Image F used in Prototype 6 both have only the possible stance value of *Reject*, through a *Diverging* and an *Additive* relationship, respectively. Therefore, by pairing all the texts from Prototype 3 with all the images from Prototype 2, we obtain 1,875 synthetic examples with a multimodal stance *Reject*. Similarly, if we pair all the texts from Prototypes 3 with all the images from the examples of Prototype 6 we obtain 2,758 synthetic examples with a multimodal stance *Reject*. In this way, we have created 4,633 synthetic examples with a multimodal stance *Reject*, which we refer to as $S_{Reject}$. Finally, if we pair all texts from Prototypes 1, 3, and 5 with all the images from the examples of Prototype 5 we can produce 29,870 synthetic examples with unchanging multimodal stance, as Image E will have *No Stance* and not modify the *Accept*, *Reject*, and *No Stance* stance values of the texts respectively. We refer to this collection of synthetic, unchanged multimodal stance examples as $S_{Same}$. In total, by inferring the text/image relations and the resulting possible stance of the tweet images, we generated 46,606 synthetic examples $S_{All}$ for training multimodal stance detection. Synthetic examples $S_{All}$

were created using the gold annotated training data from MMVAX-STANCE, which we will refer to as $D_T$. BridgeTower (BT) was selected as the multimodal architecture to train with these synthetic examples, as initial experiments demonstrated it performed best across the multimodal models on stance detection. Hyperparameters for each system are provided in detail in Appendix B.

## 5 Results and Discussion

To evaluate the quality of stance identification on the test collection from MMVAX-STANCE we use the Precision (P), Recall (R), and $F_1$ metrics for detecting the *Accept* and *Reject* values of stance. We also compute a Macro averaged Precision, Recall, and $F_1$ score. The evaluation results for MMVAX-STANCE are listed in Table 4. The bolded numbers represent the best results obtained.

Among all text-based baseline systems, the LACRscore system achieved the highest Macro $F_1$ score, demonstrating the value of incorporating attitude consistency between tweets. Image-informed text-based systems demonstrated a clear pattern: OCR did not improve the $F_1$ scores of text-based systems, while image captioning increased the Macro $F_1$ scores by 3 to 4 points over the non-caption-informed systems. Image-text aligned systems illustrated a clear performance benefit when incorporating image data directly along with text

data for stance identification, with the CLIP-Joint system achieving an improved Macro $F_1$ score over the best text-based systems. Multimodal models further demonstrated this phenomenon, with VILT, FLAVA, and BT systems matching or exceeding all text-based systems. BT achieved remarkable improvements in Macro $F_1$ score, increasing 8.5 points above the best text-based system, and increasing 6.1 points above the CLIP-Joint system. BT + $S_{All}$ - $D_T$ immediately demonstrates the value of introducing synthetic multimodal examples, as this system was not trained on any of the original training examples from MMVAX-STANCE, only the synthetic examples constructed in Section 4. The performance of BT along with training on each set of synthetic examples is provided to demonstrate how each type of synthetic example contributes to improved stance detection. BT + $S_{Same}$ shows moderate improvements in $F_1$ score across both the *Accept* and *Reject* stance values by providing more examples to learn from where the image does not contain stance information. BT + $S_{Accept}$ demonstrates a significant increase in the *Accept* $F_1$ score, as these synthetic examples provide additional opportunities for BT to identify the *Accept* stance in both the text and the images in a tweet. Similarly, BT + $S_{Reject}$ demonstrates a significant increase in the *Reject* $F_1$ score, which has a large impact on the Macro $F_1$ score, as the *Reject* stance tends to be more difficult to identify by stance detection systems. Finally, BT + $S_{All}$ incorporates all synthetic examples, providing significantly more silver examples for BT to use to identify all stance values. BT + $S_{All}$ achieves the best Macro $F_1$ score of all systems, with a score of 71.32.

| System | Synthetic Size | Macro $F_1$ |
|---|---|---|
| BT | 0 | 52.91 |
| BT + $S_{10\%}$ | 4,660 | 59.73 |
| BT + $S_{20\%}$ | 9,321 | 64.10 |
| BT + $S_{30\%}$ | 13,981 | 65.56 |
| BT + $S_{40\%}$ | 18,642 | 67.90 |
| BT + $S_{50\%}$ | 23,303 | 68.78 |
| BT + $S_{60\%}$ | 27,963 | 69.30 |
| BT + $S_{70\%}$ | 32,624 | 69.64 |
| BT + $S_{80\%}$ | 37,284 | 70.42 |
| BT + $S_{90\%}$ | 41,945 | 70.42 |
| BT + $S_{All}$ | 46,606 | 71.32 |

Table 5: Stance detection ablation experiments with BridgeTower on the test collection from MMVAX-STANCE over the size of the synthetic dataset.

We also perform an ablation experiment over the benefits of the size of the synthetic dataset. We train BridgeTower on various randomly shuffled proportions of the full synthetic dataset $S_{All}$, from 10%, corresponding to BT + $S_{10\%}$, to 90%, corresponding to BT + $S_{90\%}$. Furthermore, we compare these systems with BridgeTower trained on zero synthetic examples, corresponding to BT, and BridgeTower trained on all the synthetic examples, BT + $S_{All}$, the same systems reported in Table 4. Results of this ablation experiment are provided in Table 5. These results are not surprising, showing that by adding even only 10% of the augmented data, the Macro $F_1$-score increases from 52.91 to 59.73. This trend continues until 80% of the augmented data is employed, when the Macro $F_1$-score reaches values of over 70%.

## 6 Related Work

Several stance detection datasets using social media postings are available, all of them considering only the text of the postings. Very well-known is the stance detection dataset shared publicly with SemEval2016 Task 6 (Mohammad et al., 2016). This dataset consists of 4,163 tweets considered politics-relevant targets (e.g., Hillary Clinton) when annotating the stance of a tweet's text. A newer and much larger stance-annotated dataset (approximately 50,000 tweets), introduced in Conforti et al. (2020), used targets focused on financial transactions that involve mergers and acquisitions. Stance-annotated Twitter datasets covering topics related to the recent COVID-19 pandemic are also currently available. For example, Mutlu et al. (2020) published COVID-CQ, a dataset of approximately 14,000 tweets manually annotated to capture each tweet author's stance regarding the use of hydroxychloroquine in the treatment of COVID-19 patients. A dataset of 1,629 tweets annotated with stance was also introduced in Miao et al. (2020), targeting lockdown regulations in New York City. In Glandt et al. (2021) a stance-annotated dataset of 7,122 tweets focused on 4 targets related to the pandemic (e.g. "Wearing a Face Mask"). The COVIDLIES dataset (Hossain et al., 2020) of 6,761 stance-annotated tweets targeted 86 common misconceptions about COVID-19. Additionally, the COV-AXLIES dataset (Weinzierl and Harabagiu, 2022a) is a stance-annotated dataset of 7,152 tweets focusing on 47 frames of misinformation targeting the COVID-19 vaccines, which was expanded into the COVAXFRAMES dataset (Weinzierl and Harabagiu,

2022c), containing 14,180 stance-annotated tweets focusing on 113 COVID-19 vaccine hesitancy framings. All these datasets ignored the presence of images in tweets when judging the stance values. Although the multimedia stance annotations provided by MMVax-Stance consider only 11,300 tweets, the synthetic examples that we have generated provide significantly more examples.

Previous stance identification on Twitter was cast either as (1) a classification problem, learning to predict the stance value of a tweet towards a given target; or (2) an inference problem, when a tweet may entail, contradict, or does not imply the target. *Classification methods:* Neural architectures, relying on RNNs, CNNs, and/or attention mechanisms, dominate these methods (Augenstein et al., 2016; Du et al., 2017; Sun et al., 2018; Siddiqua et al., 2019). BERT was later shown to be the best model overall for stance detection on the SemEval2016 Task 6 (Ghosh et al., 2019). On the COVIDLIES dataset, Weinzierl et al. (2021) demonstrated that BERT can be further improved when using Graphic Attention Networks (Veličković et al., 2018). Further improvements were reported in Weinzierl and Harabagiu (2022a) when operating on CO-VAXLIES, by considering the implementation of pragmatics properties of stance into knowledge embeddings.
*Inference Methods:* When the COVIDLIES dataset of stance annotations was released in Hossain et al. (2020), stance identification was presented as a natural language inference problem, which can benefit from existing textual inference datasets. Bidirectional LSTM encoders and Sentence-BERT (SBERT) (Reimers and Gurevych, 2019) were trained on three common NLI datasets: SNLI (Bowman et al., 2015), MultiNLI (Williams et al., 2018), and MedNLI (Williams et al., 2018), helping to identify the stance of tweets. However, classification methods were able to obtain superior results on stance detection. Nevertheless, none of these methods consider the information conveyed by images present in tweets for stance detection, which our experiments on MMVax-Stance demonstrate.

# 7 Conclusion

In this paper we introduce MMVax-Stance, the first multimodal dataset of stance-annotated tweets towards 113 frames of communication. These frames communicate various aspects of COVID-19 vaccination, informed by the 7C model (Geiger

et al., 2022). Baselines were established using several text-based and multimodal stance detection systems. Substantial interactions between the text and the image(s) of a tweet were found to be key when inferring the tweet author's stance towards a frame of communication. The inference of the text/image relations led to the generation of a large set of synthetic examples of multimodal tweets, annotated with stance values. Baseline methods were greatly improved when the synthetic examples were employed, indicating the role of massive sets of examples and relations between text and images when learning to identify the values of stance in multimodal documents.

# 8 Limitations

Stance detection methods utilized in this paper focus on social media posts from Twitter. Therefore, these methods may not work as well on posts found on other platforms, especially those allowing longer content, such as Reddit. We plan to extend the methods presented in this paper to additional social media platforms in future work.

Additionally, a central assumption of our approach is the idea that relationships exist between the text of a post and the image(s) in the post. We discovered these relationships utilizing a combination of annotations and text-based stance predictions, but either of these stance values could be incorrect due to mislabeling or mistaken predictions, leading to incorrectly inferred relations. These cases certainly exist, as some of our annotated multimodal stance values contradicted the predicted text stance values. We had 526 instances of posts with an annotated multimodal stance of *No Stance* with a predicted text stance of *Accept*, and 279 instances of the predicted text stance of *Reject* with an annotated multimodal stance of *No Stance*. These cases represent 15% of the training dataset of MMVax-Stance, which aligns with expectations from the combined annotator disagreement and the accuracy of the text-based stance detector.

Finally, we focus heavily on the interactions between *parallel* and *additive* relations between the text and the image(s) of a post. Further work could expand on finding ways to also utilize the *diverging* relations, and their interactions with *parallel* and *additive* relations.

## 9 Ethics Statement

We respected the privacy and honored the confidentiality of the users that have produced the tweets pertaining to MMVAX-STANCE. We received approval from the Institutional Review Board at The University of Texas at Dallas for building and working with this Twitter social media dataset. IRB-21-515 stipulated that our research met the criteria for exemption #8(iii) of Chapter 45 of Federal Regulations Part 46.101.(b). Experiments were performed with high professional standards, avoiding evaluation on the test collection until a final model was selected from development performance. All experimental settings, configurations, and procedures were clearly laid out in this work, the supplemental material, and the linked GitHub repository. The public good was the central concern during all enclosed research, with a primary goal of benefiting both public health and natural language processing research.

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

## A Annotation Questionnaire

Annotators were first presented with the following set of instructions:

> Choose the stance value that is expressed by the tweet towards the frame text. Consider both the text of the tweet along with the image. If the text and the image are in conflict, choose the stance that

| System | Learning Rate | Batch Size | Epochs |
|---|---|---|---|
| BERT | 1e-4 | 32 | 10 |
| LES-GAT-MF | 5e-4 | 32 | 10 |
| LACRscore | 1e-4 | 32 | 36 |
| CLIP-Text | 3e-5 | 32 | 10 |
| CLIP-Image | 3e-5 | 32 | 10 |
| CLIP-Joint | 3e-5 | 32 | 10 |
| VILT | 3e-5 | 32 | 10 |
| FLAVA | 3e-5 | 16 | 100 |
| BT | 3e-5 | 32 | 10 |
| BT + $S_{All}$ - $D_T$ | 3e-5 | 64 | 1 |
| BT + $S_{Same}$ | 3e-5 | 64 | 1 |
| BT + $S_{Accept}$ | 3e-5 | 64 | 1 |
| BT + $S_{Reject}$ | 3e-5 | 64 | 1 |
| BT + $S_{All}$ | 3e-5 | 64 | 1 |

Table 6: Hyperparameters for stance detection systems on MMVAX-STANCE.

> is expressed by the text. Do not follow links in the tweet, as the stance should be determined by the text and the image of the tweet.
>
> **Accept** The user accepts, agrees with, and/or propagates the provided frame.
>
> **Reject** The user rejects, disagrees with, rebuts, debunks, or demonstrates the opposite stance towards the frame.
>
> **No Stance** The user does not demonstrate acceptance or rejection towards the frame, but the tweet is relevant to or is about the frame.
>
> **Not Relevant** The user's tweet has nothing to do with the frame, and was incorrectly predicted to be relevant by our retrieval system.

Annotators were next provided a series of examples of each stance value towards a frame.

Each example to be annotated was provided to the annotators in the following format:

> Frame Text:
> *frame*
>
> Tweet Text:
> *text*
>
> *image*

Annotators were then asked to select one of the four possible stance values: *Accept*, *Reject*, *No Stance*, or *Not Relevant*.

## B  Training Details

Primary hyperparameters for each stance detection method are provided in Table 6. All systems utilize the same learning rate schedule, where the learning rate starts at the maximum learning rate provided in Table 6, after which the learning rate is decayed to 0 over the remainder of the training steps. Any additional hyperparameters follow training hyperparameters selected in prior work, such as the size of the TransMS knowledge embeddings (8) or the margin used (4.0) by the LACRscore systems. We always attempted to train with the largest batch size (as a power of 2) which could fit on our Nvidia Titan V GPU. Hyperparameter decisions were entirely guided by maximizing $F_1$ score on the development collections of MMVAX-STANCE, and only 5 to 10 experiments with different hyperparameters were performed before the best model configurations were selected to be evaluated on the test collection. Any additional details and code can be found in the linked GitHub repository.