# OpenReview forum: "Identification of Multimodal Stance Towards Frames of Communication"
_EMNLP/2023/Conference — EMNLP 2023 Main_

### Official Review · Reviewer_Sfdh · 2023-08-03

**Soundness:** 1

**Ethical Concerns:**

Yes

**Excitement:**

1: Poor: I cannot identify the contributions of this paper, or I believe the claims are not sufficiently backed up by evidence. I would fight to have it rejected.

**Paper Topic And Main Contributions:**

The paper first introduces a new dataset of 11,300 multimodal tweets with stance annotations towards 113 different frames of communication related to COVID-19 vaccination. On this basis, the paper evaluates some baseline models on the dataset and proposes a data augmentation approach that generates 46,606 more synthetic training samples.

**Questions For The Authors:**

1. Why count "Not Relevant" tweets (according to Appendix A) into the total number of the training/development/test collection? It makes the quantitative relations in Table 2 confusing (for example, 3,145 + 1,685 + 2,832 = 7,662 ≠ 11,300).

2. Since it is necessary to "consider both the text of the tweet along with the image" when annotating the multimodal stances according to Appendix A, then why not annotate the text-based and image-based stances at the same time instead of using the model-predicted text-based stances when generating synthetic samples? It costs nearly no more effort to do this and if this is done, all the incorrectly predicted relations (such as contradictory prototypes 2 and 4) could be avoided.


**Reasons To Accept:**

The paper annotates a multimodal dataset for the stance detection task towards communication frame .

**Reasons To Reject:**

1. The paper does not clarify the concept, application scenarios, and related works of "communication frames" clearly. Is stance detection towards communication frames a new task? Is there any advantage of detecting the stance towards a specific communication frame rather than a traditional target? These questions should be answered in the paper.

2. The paper only evaluates some existing baselines and does not propose any new approach for the stance detection task.

3. The proposed data augmentation approach is intuitive, but not very novel.

4. Is it necessary to present Text-Image Relations? Are there some experiments to support your claim?

5. This paper poses a great reading challenge. Writing needs to be greatly improved.


**Reproducibility:**

1: Could not reproduce the results here no matter how hard they tried.

**Reviewer Confidence:**

4: Quite sure. I tried to check the important points carefully. It's unlikely, though conceivable, that I missed something that should affect my ratings.

---

> ### Author Rebuttal · Authors · 2023-08-28
>
> Addressing concerns and questions given for rejecting the paper:
>
> 1. Regarding the need to clarify the concept of “communication frames” – we start the introduction with a definition of these frames (lines 28-30) and a citation of an influential paper that is discussing them. Then in lines 037-041 we state why they are important- as they help people understand issues and more importantly – form their opinions! Therefore, the direct application is inferring attitudes and understanding the causes of those attitudes. In the lines 074-080 we state clearly that it is a new task – stance detection towards frames – and we explain its advantages when compared with stance towards a target, i.e. “the stance is judged against explicit aspects of the frame, not against some unspecified properties of a target." The direct advantage of using stance towards frames is in the clarity of the judgement – when deciding the stance towards a target, it is not clear what exactly about that target makes a judge agree or disagree with the target.
>
> 2. We present an empirical method that is based on data augmentation obtained by (1) considering the possible relations between texts and images and (2) the results of text-only stance detection.
>
> This data augmentation is new. Even if this method is simple, it proved to be performing very well. When state-of-the-art multimodal processing techniques were used (e.g. BridgeTower) their results alone were not very impressive, whereas when the augmented data was used with BridgeTower – a significant increase in the quality of results was obtained (see Table 3).
>
> 3. As intuitive as this approach seems after reading our paper, this approach is entirely novel and was not obvious to others in the field. We feel that the fact that it leads to a significant increase in performance over state-of-the-art methods (BridgeTower) justifies the need to report the methodology, as it may inspire others in the field. Moreover, performing multimodal stance detection with the same level of performance as text-only stance detection (over .70 F1-score) is not trivial.
>
> 4. Without considering the text-image relations, the data augmentation would not be possible. Therefore, we would only have the current state-of-the-art results generated by BridgeTower. Our simple, empirical method provides new state-of-the-art results. All experimental results listed in Table 3 support this claim.
>
> Answering the questions:
>
> 1. We count “Not Relevant” tweets in the total size of the collection because some tweets do not evoke any frame of communication, therefore they are not considered relevant to any frame. These evocation judgements were made by researchers with expertise in the Theory of Communication and we believe that these judgments should not be discarded, as they reflect the fact that people do not form attitudes based on the content of each tweet, but rather based on the content of some tweets, which happen to evoke frames. Regarding Table 2, we would be glad to include an additional column for the number of “Not Relevant” judgements in each dataset split such that these numbers are less confusing to readers.
>
> 2. There are several reasons that justify the annotation of the entire tweet’s stance – considering both the text and the image. First, as shown in Figure 1, tweet authors do not decide to use both images and texts by accident. When making that decision, they rely on  the possible relations between text and images to convey their message. This motivates our decision to ask a human judge to infer the stance of an entire tweet towards a frame, and not only of the text or image part. Secondly, while text-based stance detection was studied before, image-only stance detection suffers from the quality limitations of image processing – highlighted in the Table 3 when using CLIP-Image. Third, asking annotators to make stance judgements for the entire tweets (considering both the text and the image) and then also asking them to make stance judgements based only on the text, and finally make judgements only based on the image will cost three times more than the cost of asking them to make stance judgements based on all content of each tweet. For this annotation to be fair, annotators would need to independently judge these stance values in isolation, otherwise the order of presenting the text and image would bias annotation efforts. Moreover, when we designed the annotation process, it was not obvious that the stance values of the text and image in isolation would lead to improved results of multi-modal stance detection. Finally, by having gold-standard annotations of the text-only stance values and of the image-only stance values, the data augmentation technique would be the same. The text-based detection method might have some slight improvements, because it is fine-tuned to the gold -data, which would also lead to some slight improvements on the multi-modal stance detection. But it will not be of the same impact that we have seen from current SOTA results (with BridgeTower) to the results of using this data -augmentation on silver data.
>
> 3. Prototypes 2 and 4 are not incorrectly predicted relations, as we clearly outline in Section 4.2. Diverging relations (covered by these 2 prototypes) are a key type of relation between texts and images, which exploit to produce synthetic examples.
>
> Additionally, we would like to ask what the reviewer felt was ethically dubious about our work, as they have marked “Ethical Concerns” as “Yes.” We clearly state that we received approval from our Institution Review Board (IRB). Please clarify your ethical concerns.
>
> Finally, we would like to question the reviewer’s issue with reproducibility: we have provided an anonymized code repository along with a commitment to release annotations. Why does the reviewer feel they would never be able to reproduce our results? Please clarify your reproducibility concerns.

---

### Official Review · Reviewer_GuKD · 2023-08-03

**Soundness:** 3

**Excitement:**

3: Ambivalent: It has merits (e.g., it reports state-of-the-art results, the idea is nice), but there are key weaknesses (e.g., it describes incremental work), and it can significantly benefit from another round of revision. However, I won't object to accepting it if my co-reviewers champion it.

**Paper Topic And Main Contributions:**

This paper addresses detecting multimodal stance towards frames of communication. Previous work most predicts the stance from the text. However, social media users tend to combine images and texts to convey their stances. Therefore, the authors consider 113 frames of communication and use the frames to retrieve relevant multimodal tweets for constructing the first COVID-19 multimodal stance detection dataset. Besides, authors also utilize text/image relations to generate synthetic multimedia examples. Experimental results on the dataset show the effectiveness of the designed method for synthesizing data.

**Questions For The Authors:**

A. Could you explore the influence of synthetic data size (e.g., 10%, 40%, 70%) on the results?

**Reasons To Accept:**

1. Previous work only relies on text to detect the stance. The authors collect a multimedia stance detection dataset towards 113 frames of communication about various aspects of COVID-19 vaccination, which might promote the development of related research.
2. Many unimodal and multimodal models are utilized to perform stance prediction. Experimental results on the constructed dataset reveal the effectiveness of the combination of texts and images.


**Reasons To Reject:**

1. The authors select the tweets which are relevant to the frame of communication when collecting the data. However, this might result in a lack of diversity in the collected data. Besides, they don’t conduct the deduplication for the retrieved similar data. Actually, the same multimodal tweets might be contained in the retrieved data.

2. The method based on text-image relations for generating synthetic multimedia stance data is specially designed and somewhat complex. It might be better if you could use other generic and simple methods (e.g., self-training) to synthesize data.


**Reproducibility:**

4: Could mostly reproduce the results, but there may be some variation because of sample variance or minor variations in their interpretation of the protocol or method.

**Reviewer Confidence:**

4: Quite sure. I tried to check the important points carefully. It's unlikely, though conceivable, that I missed something that should affect my ratings.

---

> ### Author Rebuttal · Authors · 2023-08-28
>
> We appreciate the reviewer recognizing that a multimedia stance detection dataset could promote the development of related research. Addressing concerns and questions given for rejecting the paper:
>
> 1. Lines 184-192 describe how we performed de-duplication, therefore there are no duplicates in the dataset. When selecting tweets relevant to a frame of communication, we also took into account the fact that a tweet may evoke more than one frame, therefore the tweets may be relevant to more than one frame. Figure 2 illustrates the property of frames, which cannot be ignored. However, we made sure that if a tweet evokes more than one frame, it should be used either only in the training dataset of all the frames it evokes, or it should be used only in the test dataset of those frames, or in the development dataset of all the frames. Therefore, no duplicates exist in evaluations. We would be glad to report the number of unique tweets, in contrast to tweet judgements, in each split: 4,882 in training, 646 in development, and 1,390 in testing.
>
> 2. As shown in Table 3, we have considered 18 simpler methods, which were applied on our dataset. The results of these simpler methods did not obtain better scores than 52.91 F1-score. By using the data augmentation methods that we present in this paper as well as a text-based stance detection we were able to obtain an augmentation of the values of the F1-score by nearly 20 points, which we feel is significant. Although the data augmentation method may seem complicated, it makes it possible to obtain results for multimodal stance detection that are comparable with text-only stance detection results. We have also experimented with self-training and the results were poor, therefore we did not feel that they are worth reporting. In lines 330-344 we mention relations between texts and images in general citing two books that detail them, which all multimodal data exhibits– there is nothing specially designed about them.
>
> 3. We agree that an ablation study over the synthetic data size would be insightful, and we would be glad to add these results to the paper:
>
> 0%: 52.91 F1 (BridgeTower baseline)
>
> 10%: 59.73 F1
>
> 20%: 64.10 F1
>
> 30%: 65.56 F1
>
> 40%: 67.90 F1
>
> 50%: 68.78 F1
>
> 60%: 69.30 F1
>
> 70%: 69.64 F1
>
> 80%: 70.42 F1
>
> 90%: 70.29 F1
>
> 100%: 71.32 F1 (BT-ALL)
>
> These results are not surprising, showing that by adding even only 10% of the augmented data, the F1-score increases from 52.91 to 59.73. This trend continues until 80% of the augmented data is used, when the F1-score reaches values of over 70%.

---

### Official Review · Reviewer_anjp · 2023-08-05

**Soundness:** 3

**Excitement:**

3: Ambivalent: It has merits (e.g., it reports state-of-the-art results, the idea is nice), but there are key weaknesses (e.g., it describes incremental work), and it can significantly benefit from another round of revision. However, I won't object to accepting it if my co-reviewers champion it.

**Paper Topic And Main Contributions:**

The main contribution of this work is proposing a new multi-modal stance detection dataset, namely MMVAX-STANCE. Besides, how multi-modal information impacts the multi-modal stance detection is investigated.

**Questions For The Authors:**

A: See "Reasons To Reject 2".

**Reasons To Accept:**

1. This work is well motivated.
2. The proposed dataset has a solid theoretical background.
3. Due to the annotating difficulty, the work of this paper is abundant.

**Reasons To Reject:**

1. The task of multi-modal stance is not novel and there are some existing relevant datasets. There is no need to define a new object "Frame of Communication".
2. There are totally 9 possible combinations of multi-modal annotations and text-based silver annotations. Are "No Stance (multi-modal anno.) - Accept (silver anno.)" and "No Stance (multi-modal anno.) - Reject (silver anno.)" not prototypical enough? Or do they just not exist at all?

**Reproducibility:**

4: Could mostly reproduce the results, but there may be some variation because of sample variance or minor variations in their interpretation of the protocol or method.

**Reviewer Confidence:**

4: Quite sure. I tried to check the important points carefully. It's unlikely, though conceivable, that I missed something that should affect my ratings.

---

> ### Author Rebuttal · Authors · 2023-08-28
>
> We appreciate your recognition of the annotating difficulty behind the multimodal stance dataset proposed. Addressing concerns and questions given for rejecting the paper:
>
> 1.a. We searched the ACL anthology as well as AAAI/IJCAI papers and could not find any paper tackling multi-modal stance detection on social media. There may be many other multimodal datasets, but they are used to process other tasks – and not stance detection. If the reviewer has knowledge of such datasets, providing a reference would help us to correct our statement and link to the reference. Otherwise, we will have no reason to not continue claiming that to our knowledge this task and dataset is novel.
>
> 1.b. The new “object”, referred by the reviewer, is defined as frame of communication because such frames are well-known in the Theory of communication. Lines 028-030 of the paper – which correspond to the first sentence of the introduction, define these frames  – and cite an influential paper that introduced them. Frames of communication are important because they promote a particular interpretation and moral evaluation of problems promote a particular problem definition, causal interpretation, moral evaluation and/or treatment evaluation.  Moreover, they are ubiquitous in natural language communications, (As we elaborate in lines 037-041) therefore they merit attention from NLP research.  Previously, papers focusing on frames of communication have been accepted to ACL/EMNLP (search on aclweb anthology proves it). We feel that by not providing such a definition and discussing some undefined “objects” would make the paper ill-defined and confusing.
>
> 2.While there are 9 possible combinations of [Multimodal_Stance_VALUE, Text_Stance_VALUE] only 7 of these combinations bring any value to the data augmentation that we have performed.  The two cases which we ignored are: (Case 1) Multimodal_Stance_VALUE= “No Stance” and Text_Stance_VALUE= “Accept”; and (Case 2) Multimodal_Stance_VALUE= “No Stance” and Text_Stance_VALUE= “Reject”. Multimodal_Stance_VALUE is decided by a human judge, whereas the  Text_Stance_VALUE is decided by an automatic stance detection system. Since we always trust the human judge, which decided the stance based on both the text and the image of the tweet, we infer that in both Case 1 and Case 2, the automatic stance detection was incorrect. Therefore, we did not consider any cases when we knew that the text stance detection was incorrect. Only 15% of all multimedia tweets pertained to Case 1 or 2.
>
> Thank you very much for reviewing our work!

---

### Meta-Review · Area_Chair_MbQG · 2023-09-16

**Recommendation:** 3

**Metareview:**

This work proposes multimodal stance classification and provides a new dataset. The dataset collection procedure is well explained. However, it could benefit from more details about the annotators: who were they? How long does it take to label? How much were they paid? I would suggest statistics about the dataset's language content: #tokens, #words, etc. The proposed method is reasonable, and the experimentation is exhaustive. Listed issues have been addressed and defended in rebuttal comments, and there is openness to clarify different aspects of the final version of the manuscript. The dataset might be of interest to the community, and it would be available upon publication.

Pros
* Novel dataset and stance-classification-related task
* Exhaustive evaluation and promising results

Cons
* Missing minor dataset information about annotators and language content

---

### Decision · Program_Chairs · 2023-10-07

**Decision:**

Accept-Main

**Comment:**

This work proposes multimodal stance classification and provides a new dataset. The dataset collection procedure is well explained. However, it could benefit from more details about the annotators: who were they? How long does it take to label? How much were they paid? I would suggest statistics about the dataset's language content: #tokens, #words, etc. The proposed method is reasonable, and the experimentation is exhaustive. Listed issues have been addressed and defended in rebuttal comments, and there is openness to clarify different aspects of the final version of the manuscript. The dataset might be of interest to the community, and it would be available upon publication.

Pros
* Novel dataset and stance-classification-related task
* Exhaustive evaluation and promising results

Cons
* Missing minor dataset information about annotators and language content